# A Simple Self-labeling Method for Semi-supervised Medical Image Segmentation

Ye Zhu[1] and Hanlin Tian[1]

The Chinese University of Hong Kong (Shenzhen), China
`zhuye1@cuhk.edu.cn`

**Abstract.** Leveraging a few labeled images and a large number of unlabeled images is crucial for medical image segmentation since labeling the medical data can be very expensive and time-consumed. Therefore we introduce a naïve but simple method to utilize the massive unlabeled medical images for better training. We first use all the labeled data to train a basic model, then use this pre-trained model to infer the unlabeled images to get pseudo-labels, and finally use all the obtained pseudo-labels and the original labels as the ground truth of all images, and retrain the model from scratch to acquire the final model. We believe this is a simple but effective way to utilize the massive number of unlabeled images and experiments were performed to evaluate such method.

**Keywords:** Semi-supervise learning · Pseudo labels.

## 1 Introduction

Leveraging a few labeled images and a large number of unlabeled images is crucial for medical image segmentation since labeling the medical data can be very expensive and time-consumed. Motivated by this, many semi-supervised segmentation methods [6] were developed to exploit the information contained in unlabeled images.

Recent semi-supervised approaches in medical image segmentation are mainly relied on pseudo-labeling, contrastive learning and consistency regularization [11,12,10,13]. In [13], a cross-level contrastive algorithm is developed to enhance the representation capacity for local features in semi-supervised semantic segmentation. A self-prototype alignment is proposed to learn more stable region-wise features within unlabeled images, which can optimize the classification margin by boosting in intra-class compactness and inter-class separation on the feature space [12]. Moreover, a framework improve the accuracy of the pseudo labels using the features and edges of the superpixel maps, and achieve great performance in brain tumor region segmentation [10].

Rather than using complicated and well-designed methods, we proposed a simple strategy to use a well-trained model to generate pseudo labels for a large number of unlabeled images, and finally use all of them to retrain the model from scratch. It is a easy way to utilize the unlabeled images and achieve better performance than only using limited labeled images.

The remainder of this paper is organized as follows. We introduce the detail of the preprocessing and proposed method in Section 2. Then, the experiment details are presented in Section 3 and finally the results and discussion comes in section 4.

## 2   Method

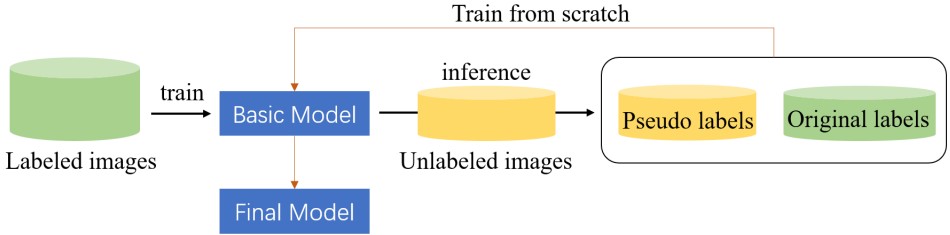

**Fig. 1.** Training strategy

The proposed method is illustrated in Figure 1.

### 2.1   Preprocessing

For data preprocessing, we followed the work in [3]. Considering the characteristics of CT images, each 3D image top 5% of its intensity histogram was cut off for alleviating artifacts. Then each 3D image was standardized and sliced to 2D images to suit the base network setup. The standardization equation can be formulated as:

$$image = (image - image.mean())/image.std() \qquad (1)$$

### 2.2   Proposed Method

In this paper, we proposed a simple but effective method based on the 2D Swin-Unet, where a U-Net architecture is adopted. Motivated by the Swin Transformer's success, the Swin-Unet leverage the power of Transformer for 2D medical image segmentation and achieve great performance.In this task, we use Swin-Unet as our basic network for semi-supervised segmentation. The network architecture is depicted in Figure 2.

Swin-Unet is a end-to-end training frameworka and is first introduced Transformer-based U-shaped architecture that consists of encoder, bottleneck, decoder, and skip connections. The input medical 2D slices are split into non-overlapping patches and each patch is treated as a token and fed into the Swin-Transformer-base encoder to acquire deep feature representations. This extracted features will

be up-sampled by the decoder and finally fused with the multi-scale features from the encoder via skip connections.

The Swin-Unet was first trained with only labeled images. The iterative process may go on until the convergence is met. Then this well-trained model is used for generating pseudo labels for the corresponding unlabeled images. After the pseudo labels are acquired, both labeled and unlabeled images are used for training from scratch, and the original and pseudo labels are served as the ground truth. In this way, we are able to take full advantage of the large number of unlabeled data.

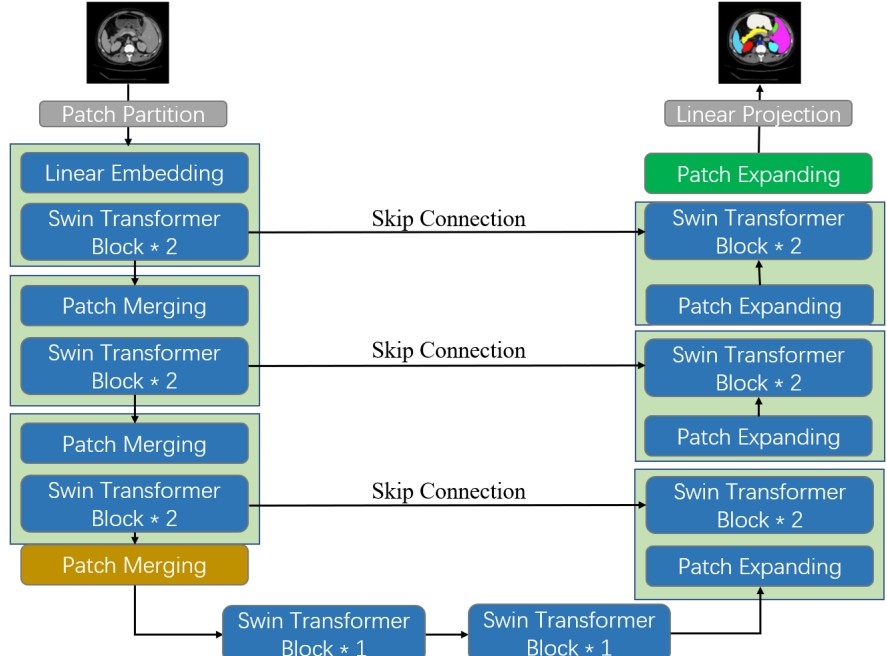

**Fig. 2.** Network architecture

The loss function we use is the summation between Dice loss and cross entropy loss, it is believed that the compound loss functions are robust in various medical image segmentation tasks [7].

## 3    Experiments

### 3.1    Dataset and evaluation measures

The FLARE2022 dataset is curated from more than 20 medical groups under the license permission, including MSD [9], KiTS [4,5], AbdomenCT-1K [8], and

TCIA [2]. The training set includes 50 labelled CT scans with pancreas disease and 2000 unlabelled CT scans with liver, kidney, spleen, or pancreas diseases. The validation set includes 50 CT scans with liver, kidney, spleen, or pancreas diseases. The testing set includes 200 CT scans where 100 cases has liver, kidney, spleen, or pancreas diseases and the other 100 cases has uterine corpus endometrial, urothelial bladder, stomach, sarcomas, or ovarian diseases. All the CT scans only have image information and the center information is not available.

The evaluation measures consist of two accuracy measures: Dice Similarity Coefficient (DSC) and Normalized Surface Dice (NSD), and three running efficiency measures: running time, area under GPU memory-time curve, and area under CPU utilization-time curve. All measures will be used to compute the ranking. Moreover, the GPU memory consumption has a 2 GB tolerance.

### 3.2 Implementation details

**Environment settings** The development environments and requirements are presented in Table 1.

**Table 1.** Development environments and requirements.

| | |
|---|---|
| Windows/Ubuntu version | Red Hat 8.5.0-10 |
| CPU | Intel(R) Xeon(R) Silver 4210R CPU @ 2.40GHz |
| RAM | 16×4GB; 2.67MT/s |
| GPU (number and type) | One NVIDIA V100 16G |
| CUDA version | 11.0 |
| Programming language | Python 3.7 |
| Deep learning framework | Pytorch (Torch 1.6.0, torchvision 0.7.0) |

**Training protocols** For data augmentation, we applied the simple operation such as random rotate and random flip. In the training phase, we randomly select 20 cases from all training cases to train our model, and the rest 30 cases are served as our validation set. The first basic model was trained for 1000 epochs, then with the massive generated pseudo labels, the second training phase was set to 50 epochs. The model was validated every epoch, then the model which has the highest DSC and NSD value is selected as the best model to inference the test set.

## 4 Results and discussion

In Table. 4 and Table. 5, the results show the effect of using unlabelled cases. The value of DSC improved from 73.9% to 79.1% and NSD improved from 80.2% to 86.0% which indicates that our method has taken advantages of using a large number of unlabeled images. But we also noticed that our method did

**Table 2.** Training protocols.

| | |
|---|---|
| Network initialization | Truncated normal initialization |
| Batch size | 18 |
| Image size | 3×224×224 |
| Total epochs | 1000 |
| Optimizer | Adam optimizer |
| Initial learning rate (lr) | 0.001 |
| Lr decay schedule | LR = baseLR*(1.0-NumOfIter/MaxIterations)**0.9 |
| Training time | 39 hours |
| Number of model parameters | 27.17M |
| Number of flops | 6.19G[1] |

**Table 3.** Training protocols for the refine model

| | |
|---|---|
| Network initialization | Truncated normal initialization |
| Batch size | 18 |
| Patch size | 3×224×224 |
| Total epochs | 100 |
| Optimizer | Adam optimizer |
| Lr decay schedule | LR = baseLR*(1.0-NumOfIter/MaxIterations)**0.9 |
| Training time | 60 hours |
| Number of model parameters | 27.17M |
| Number of flops | 6.19G[2] |

**Table 4.** Comparisons of our full model with previous model only trained with labeled data with respect to DSC accuracy metric. The results are coming from our divisions of training-validation sets (20-30 from all labeled cases).

| Methods | Labels | Unlabels | Liver | Rkidney | Spleen | Pancreas | Aorta | IVC | RAG | LAG | Gallbladder | Esophagus | Stomach | Duodenum | Lkidney | Mean DSC |
|---|---|---|---|---|---|---|---|---|---|---|---|---|---|---|---|---|
| Swin-Unet [1] | 20 | 0 | 93.6% | 80.9% | 89.8% | 58.7% | 86.0% | 77.7% | 63.3% | 56.1% | 73.7% | 70.4% | 78.2% | 58.3% | 75.0% | 73.9% |
| Ours | 20 | 30 | 95.3% | 90.7% | 92.7% | 64.9% | 87.7% | 79.7% | 66.0% | 66.0% | 77.2% | 74.1% | 82.5% | 61.9% | 89.5% | 79.1% |

**Table 5.** Comparisons of our full model with previous model only trained with labeled data with respect to NSD accuracy metric. The results are coming from our divisions of training-validation sets (20-30 from all labeled cases).

| Methods | Labels | Unlabels | Liver | Rkidney | Spleen | Pancreas | Aorta | IVC | RAG | LAG | Gallbladder | Esophagus | Stomach | Duodenum | Lkidney | Mean NSD |
|---|---|---|---|---|---|---|---|---|---|---|---|---|---|---|---|---|
| Swin-Unet [1] | 20 | 0 | 91.5% | 73.5% | 86.5% | 79.3% | 83.2% | 65.6% | 83.4% | 72.0% | 75.3% | 89.0% | 83.9% | 84.6% | 74.6% | 80.2% |
| Ours | 20 | 30 | 95.3% | 88.5% | 91.8% | 81.4% | 86.6% | 73.3% | 84.9% | 81.0% | 81.0% | 91.4% | 86.8% | 88.2% | 88.2% | 86.0% |

**Table 6.** Comparisons of our full model with previous model only trained with labeled data with respect to accuracy metric. The results are obtained from the official validation set of FLARE2022.

| Methods | Labels | Unlabels | Liver | Rkidney | Spleen | Pancreas | Aorta | IVC | RAG | LAG | Gallbladder | Esophagus | Stomach | Duodenum | Lkidney | Mean DSC |
|---|---|---|---|---|---|---|---|---|---|---|---|---|---|---|---|---|
| Swin-Unet [1] | 20 | 0 | 70.9% | 38.2% | 63.0% | 30.8% | 60.6% | 43.7% | 22.5% | 22.2% | 35.1% | 41.67% | 32.3% | 24.0% | 41.9% | 40.5% |
| Ours | 20 | 30 | 70.6% | 52.3% | 66.0% | 38.3% | 64.1% | 49.4% | 29.1% | 29.7% | 38.7% | 49.6% | 40.4% | 26.0% | 48.1% | 46.3% |

not performed well in the official test set from FLARE22 challenge. We believed that the main reason is because our model was trained with a small amount of labeled data causing overfitting to training set.

### 4.1    Segmentation efficiency results

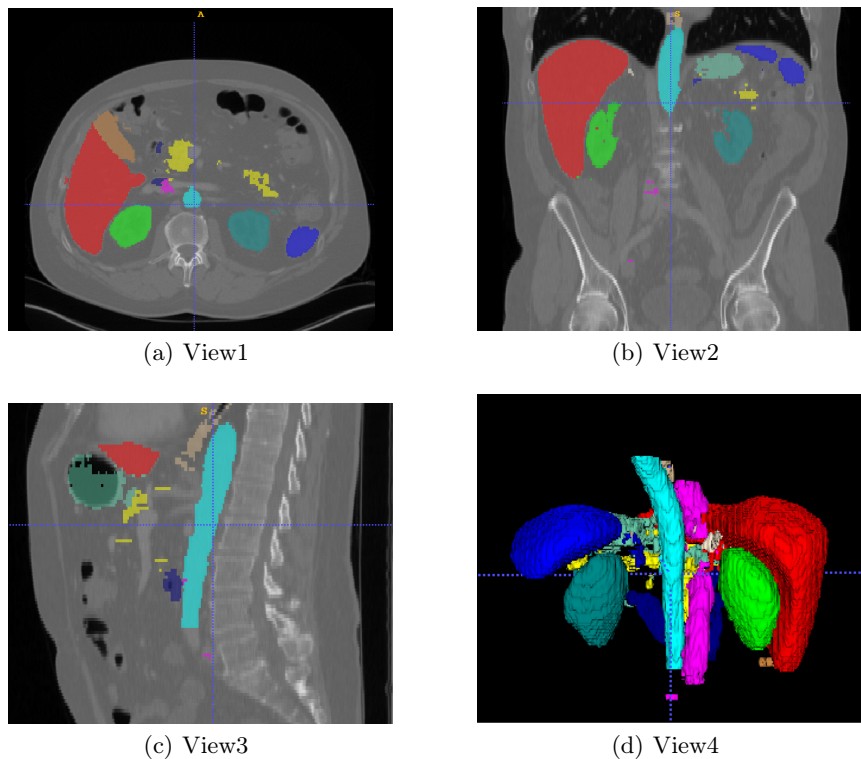

(a) View1                                   (b) View2

(c) View3                                   (d) View4

**Fig. 3.** Different views of pseudo labels

## 5    Conclusion

In this paper, we introduce a naïve but simple method to utilize the massive unlabeled medical images for better training. From the experiments results we found that this simple but effective method can improve the performance compared with using only labeled images. However, this method highly relies on the quality of the pseudo labels, and it is difficult for this self-labeling strategy to rectify the incorrect predictions. In the future work, we will focus on generating more accurate pseudo label for retraining the model.

**Acknowledgements** The authors of this paper declare that the segmentation method they implemented for participation in the FLARE 2022 challenge has not used any pre-trained models nor additional datasets other than those provided by the organizers. The proposed solution is fully automatic without any manual intervention.

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
