# OpenReview forum: "A Simple Self-labeling Method for Semi-supervised Medical Image Segmentation"
_MICCAI.org/2022/Challenge/FLARE_

### Official Review · Reviewer_Z67J · 2022-09-12
**This paper has sufficient sections, these beginning sections are good but the result section does not have enough detailed analysis**

**Rating:** 4
**Confidence:** 4

**Review:**

Authors adopt Swin U-Net as 2D segmentation problem, where they first train on labeled data, then use that trained model to generate pseudo-labels, then retrain on all of them.

This paper has sufficient sections, those beginning sections are good whereas the result section does not have enough detailed analysis; moreover the Mean DiceScore is kinda low.

Suggested improvements:
- More comments should be given on qualitative and quantitative results (such as comparison why a method performs better than others)
- Table 4, 5, 6 are difficult to read. They can be written vertically with newlines.
- Figure 3 visualization is confusing, should follow settings that are recommended by the organizer.
- An unclear point: Why use only 30 unlabeled cases for the official validation set ? Did you use 2000 the provided unlabeled cases ?

---

### Official Review · Reviewer_sdCg · 2022-09-13
**Unclear description of the method**

**Rating:** 5
**Confidence:** 4

**Review:**

1、The description of the method is not clear enough for the network structure and training parameters.
2、The authors need to highlight their contributions and results.
3、Still some mistakes in English writing.

---

### Official Review · Reviewer_soN3 · 2022-09-14
**the authors' results have room for further improvement.**

**Rating:** 5
**Confidence:** 4

**Review:**

Comments to the Author

In this paper, building on previous work, the authors apply a naive but simple method to utilize the massive unlabeled medical images for better training. But I found that for the task of this paper, the authors' results have room for further improvement.

Some examples of errors in language and figures:

- In order for the reader to be able to see the author's work in advance, it is necessary to state the results in the abstract.
- Section 2.2, paragraph 2, first line misspelled "frameworka".
- The title of Fig. 2 should be described in more detail so that the reader can know what is described even without looking at the main text.
- Table 4, Table 5 and Table 6 are not convenient for readers to read. A better way is to change the typesetting and text direction.
- etc..

Please go through the paper and improve the experimental results and wording.

---

### Official Review · Reviewer_HCNH · 2022-09-15
**The authors propose a self-labeling method that trains a Swin-Unet for segmenting the test set**

**Rating:** 5
**Confidence:** 3

**Review:**

Strengths: Simple and easy to implement method.
Weaknesses: The proposed method should be described in better detail. There is lack of discussion of  the model and efficiency results.
Details:
- The metrics in the validation set are not included in the abstract
- Grammatical and spelling mistakes.
- There is no explanation of Figure 1.
- Figure 3 is just included in the manuscript without a proper introduction of description.
- There is no efficiency discussion.

---

### Official Review · Reviewer_2vrT · 2022-09-19
**Decent approach to self-training based on Swin-Unet, however missing important details.**

**Rating:** 5
**Confidence:** 3

**Review:**

This paper combines a self-labeling approach with a 2D Swin-Unet model. In the first stage, a 2D Swin-Unet is trained on a subset of 20 labeled training images. This network is then used to generate pseudo labels on the unlabeled data. In stage 2, a newly initialized model is then trained on the union of labeled and pseudo labeled images. The best-performing state of the final network is used for inference on the test set.

Pros:
- Figure 1 is a simple, yet clear description of the proposed methodology.
- Figure 2 is a nice overview of the Swin-Unet network architecture, including the patch merging.

Missing details:
- Missing info on inference requirements in terms of GPU memory, speed, etc. as required by organizer's checklist.
- Details on the exact Swin-Unet architecture are missing. Input patch size, etc.

Problems:
- Typo in abstract: time-consumed -> time-consuming (minor).
- Typo in second paragraph of section 2.2: frameworka -> framework (minor).
- Information on number of epochs in training protocol assumes 1 epoch = 1 pass over whole dataset definition, but doesn't mention it (minor).
- There is a very large performance drop on the validation set. This requires investigation. The authors hint at reasons as to why this might be the case; however, the paper seems incomplete without more investigations on this.

---

### Official Review · Reviewer_x9iw · 2022-09-20
**Good try but the manuscript is incomplete**

**Rating:** 4
**Confidence:** 3

**Review:**

 In this paper, the authors developed a semi-supervised medical image segmentation model by using the pseudo-label method.

Suggestions or deficiencies:

1. The authors should add strategies to improve inference speed and reduce resource consumption and a clear description of post-processing in section 2.
2. In Table 2 and Table 3, the authors should add the loss function in the next row after Training time.
3. The authors should visualize some good cases and bad cases, then analyze the reasons for the appearance of good cases and bad cases in section 4.
4. The authors should add segmentation efficiency results and segmentation efficiency analysis in section 4.
5. The authors should add a subsection of "Limitation and future work" in section 4.

---

### Official Review · Reviewer_46nG · 2022-09-21
**A simple but effective method is proposed based on the 2D Swin Unet, where a U-Net architecture is adopted. However, a weak paper.**

**Rating:** 6
**Confidence:** 3

**Review:**

In this manuscript, the authors used all the labeled data of FLARE2022 to train a basic model, further followed by the use of the pre-trained model to infer the unlabeled images to get pseudo-labels, and finally, used all the obtained pseudo-labels and the original labels as the ground truth of all images, and retrain the model from scratch to acquire the final model.

Strengths: A simple but effective method is proposed based on the 2D Swin Unet, where a U-Net architecture is adopted.

Weaknesses: Engineering method

Suggestions for authors:

1: At least three relevant keywords

2: It is better to mention your quantitative results (mean DSC and NSD) in the last of the abstract.

3:  Why did you follow the work of [3] for the preprocessing step, and any specific reason and effect?

4: Provide an exact reference for 2D Swin-Unet for their first appearance in the text. Likewise, also provide a reference for U-Net.
5: Provide better captions for Figure 1 and Figure 2.

6: The paper methodology is incomplete and unclear; it is suggested to add the appropriate methodology of the proposed framework.
7: Please provide specific dependencies in Table 1 (if you have any).

8: Provide loss function in Table 2.

9: Provide a little more detail for Table 4.

10: Suggested adding an ablation study.

---

### Meta-Review · Program_Chairs · 2022-09-28

**Recommendation:** Major Revision
**Confidence:** 5

**Metareview:**

Reviewers raise many concerns and suggestions. Please address all comments in the revised manuscript.